# Effects of Robot-Assisted Gait Training in Patients with Burn Injury on Lower Extremity: A Single-Blind, Randomized Controlled Trial

**DOI:** 10.3390/jcm9092813

**Published:** 2020-08-31

**Authors:** So Young Joo, Seung Yeol Lee, Yoon Soo Cho, Kuem Ju Lee, Cheong Hoon Seo

**Affiliations:** 1Department of Rehabilitation Medicine, Hangang Sacred Heart Hospital, College of Medicine, Hallym University, Seoul 07247, Korea; anyany98@naver.com (S.Y.J.); hamays@hanmail.net (Y.S.C.); 2Department of Physical Medicine and Rehabilitation, College of Medicine, Soonchunhyang University Hospital, Bucheon 14584, Korea; shouldetz@gmail.com; 3Department of Rehabilitation & Assistive Technology, Korea National Rehabilitation Research Institute, National Rehabilitation Center, Seoul 01022, Korea; kjlee74@korea.kr

**Keywords:** burns, rehabilitation, robot-assisted gait training

## Abstract

This study investigated the effects of robot-assisted gait training (RAGT) on gait function in burn patients. Briefly, 40 burn patients were randomly divided into an RAGT group or a conventional training (CON) group. SUBAR^®^ (Cretem, Korea) is a wearable robot with a footplate that simulates normal gait cycles. The RAGT group underwent 30 min of robot-assisted training using SUBAR^®^ with 30 min of conventional physiotherapy once a day, 5 days a week for 12 weeks. Patients in the CON group received 30 min of overground gait training and range-of-motion (ROM) exercises twice a day for 5 days a week for 12 weeks. The RAGT group and the CON group underwent 60 min of training per day. The intervention frequency and duration did not differ between the RAGT group and the CON group. The main outcomes were functional ambulatory category (FAC); 6-min walking test (6MWT); visual analogue scale (VAS) during gait movement; ROM; and isometric forces of bilateral hip, knee, and ankle muscles before and after 12 weeks of training. The results of the VAS, FAC, and 6MWT (8.06 ± 0.66, 1.76 ± 0.56, and 204.41 ± 85.60) before training in the RAGT group improved significantly (4.41 ± 1.18, 4.18 ± 0.39, and 298.53 ± 47.75) after training (*p* < 0.001, *p* < 0.001, and *p* < 0.001). The results of the VAS, FAC, and 6MWT (8.00 ± 1.21, 1.75 ± 0.58, and 220.94 ± 116.88) before training in the CON group improved significantly (5.00 ± 1.03, 3.81 ± 1.05, and 272.19 ± 110.14) after training (*p* < 0.001, *p* < 0.001, and *p* = 0.05). There were differences in the improvement of results of the VAS, FAC, and 6MWT between groups after training, but they were not statistically significant (*p* = 0.23, *p* = 0.14, and *p* = 0.05). The isometric strengths of the right hip extensor (*p* = 0.02), bilateral knee flexor (*p* = 0.04 in the right, and *p* = 0.001 in the left), bilateral knee extensor (*p* = 0.003 in the right, and *p* = 0.002 in the left), bilateral ankle dorsiflexor (*p* = 0.04 in the right, and *p* = 0.02 in the left), and bilateral ankle plantarflexor (*p* = 0.001 in the right, and *p* = 0.008 in the left) after training were significantly improved compared with those before training in the RAGT group. The ROMs of the right knee extension (*p* = 0.03) and bilateral ankle plantarflexion (*p* = 0.008 in the right, and *p* = 0.03 in the left) were significantly improved compared with measurements before training in the RAGT. There were no significant differences of the isometric strengths and ROMs of the bilateral hip, knee, and ankle muscles after training in the CON group. There were significant improvements in the isometric strengths of the left knee flexor (*p* = 0.01), left ankle dorsiflexor (*p* = 0.01), and left ankle plantarflexor (*p* = 0.003) between the two groups. The results suggested that RAGT is effective to facilitate early recovery of muscles strength after a burn injury. This is the first study to evaluate the effectiveness of RAGT in patients with burns compared with those receiving conventional training. The absence of complications in burn patients provides an opportunity to enlarge the application area of RAGT.

## 1. Background

During rehabilitation of patient with burns, contracture is the major complication. The factors contributing to joint contracture are long-term immobilization, multiple surgical procedures, and scar formation. Early rehabilitation after a burn injury has been recommended [1,2]. A major impairment causing gait dysfunction is the limited range of motion (ROM) in the lower extremities. The ankle was the most commonly affected joint in the lower extremity, followed by the hip and the knee [3]. Decreased ROM in the lower extremities is associated with gait disturbances, such as reduced step height and step length and decreased gait speed.

Gait disturbances are among the most disabling symptoms experienced by burn patients. Rehabilitation has been essential for improving the gait and independence. Conventional training consisted of active ROM exercises, transfer training, and weight bearing training. If the patients were able to stand, they were assisted to ambulate progressively and start overground gait training. Robot-assisted gait training (RAGT) was developed to restore gait function through repetitive training [4,5,6,7]. RAGT improved gait independence in stroke and cerebral palsy patients and improved motor function [8,9]. The investigators found the effectiveness of RAGT in patients with musculoskeletal injuries such as knee osteoarthritis [10,11]. Few previous studies have reported the use of RAGT in other musculoskeletal diseases, and there are no clear guidelines regarding the application of RAGT. In the review article on robot rehabilitation, research was being conducted through a small number of experimental groups [12].

In particular, skin injury caused by burns was known as a contraindication to robot rehabilitation. Joo et al. announced the possibility of robot training for burn patients for the first time [13]. We hypothesized that RAGT may be useful in improving the gait performance of patients with a burn injury. In the present randomized controlled trial, we aimed to verify whether RAGT could be used in the rehabilitation of burn patients to promote functional recovery.

## 2. Methods

This study was a prospective, single-blind, randomized controlled trial. Eligible participants were randomly allocated to an RAGT group or a conventional training (CON) group (Figure 1). No significant differences were observed in the baseline characteristic and values before training (Table 1). We recruited 40 patients from the Department of Rehabilitation Medicine at Hangang Sacred Heart Hospital in Korea to participate in this study, between October 2019 and August 2020. Our study was approved by the Ethics Committee of the Hangang Sacred Heart Hospital (HG2018-025). All 40 patients were provided a written informed consent thereafter. Our study was approved by ClinicalTrials (NCT04281394). Numbers were assigned to 40 burn patients who satisfied all the aforementioned criteria according to their order of admission. A computer program was used to randomly divide them into the RAGT group (*n* = 21) or the CON group (*n* = 19). Four patients in the RAGT group and three patients in the CON group dropped out of the study because they did not want to undergo serial evaluations after recovering their gait function and did not visit the outpatient clinic for gait training.

Patients who underwent split-thickness skin graft (STSG) at Hangang Sacred Heart Hospital, with full or virtually full thickness involvement (>50% of the body surface area of the lower extremity), aged >18 years, and with ≤1 functional ambulation category (FAC) score of ≤3 were included in this study (Figure 2). This study excluded patients who had fourth-degree burns (involving muscles, tendons, and bone injuries) and musculoskeletal diseases (fracture, amputation, rheumatoid arthritis, and degenerative joint diseases) involving the burned lower extremity. Patients with cognitive disorders, intellectual impairment before burn injury, serious cardiac dysfunction, body weight ≥100 kg (due to problems with the belt length for fixing the thigh and calf), severe fixed contracture, skin disorders that could be worsened by RAGT and conventional intervention, and severe pain who were unable to undergo rehabilitation programs were excluded.

SUBAR^®^ (Cretem, Korea) is a wearable robot with a footplate (Figure 3). The patient’s thigh length and lower leg length were measured before training, so that the SUBAR^®^ can be adjusted to the patient’s size to ensure accurate training. Periodic movement of the lower extremities during a gait process was simulated at a tolerable and comfortable walking speed adjusted to 0.8–2.4 km/h. The conditions of the participant’s speed, step length, and degree of knee flexion were adjusted based on three parameters. RAGT can be performed regularly over a long period. The patients underwent 30 min of robot-assisted training using SUBAR^®^ in the morning with 30 min conventional physiotherapy in the afternoon, 5 days a week for 12 weeks. The patients in the CON group received 30 min of overground gait training and ROM exercises twice a day in the morning and afternoon, for 5 days a week for 12 weeks. Both RAGT group and CON group underwent 60 min of training per day. The intervention frequency and duration did not differ between the RAGT group and the CON group. Under Korean insurance, inpatients can be given physical therapy twice a day, and the treatment cost of the two groups was the same.

Outcome measurements and data analysis were performed by a trained and blinded outcome assessor who was not involved in the intervention. The outcome measures were evaluated before training and immediately after 12 weeks of training. The primary outcome measures were the ROM in bilateral hip, knee, and ankle joint. The active ROM of different joints was measured using a goniometer following a standardized technique [14]. To evaluate functional recovery, FAC scores and 6-min walking test (6MWT) distances were measured. FAC was evaluated based on a 6-point scale: 0, the patient cannot walk or can only walk with the assistance of two people; 1, the patient can only relieve body weight and manage balance with continuous aid of one person; 2, the patient can walk with continuous or intermittent assistance of one person; 3, the patient can walk under supervision other than direct assistance; 4, the patient can walk independently on a level ground, but needs help on stairs, slope, or rough road; and 5, the patient can walk independently. 6MWT was performed in accordance with the standardized guidelines, and the walking course was 20 m long. Patients were instructed to walk as far as possible in 6 min. The visual analogue scale (VAS) was used to rate the degree of subjective pain during gait movement: 0 points were assigned when no pain was noted, and unbearable pain was assigned 10 points. VAS in gait was measured using a questionnaire. Isometric muscle strengths of the hip extensors, hip flexors, knee extensors, knee flexors, ankle dorsiflexors, and ankle plantar flexors were measured using a handheld dynamometer. Two measurements were taken using a handheld dynamometer [15]. Isometric muscle strength was measured using the MicroFET II^TM^ (Hoggan Health Industries, Draper, UT, USA) (Figure 4). Each trial lasted for 3–5 s, with a 30-s rest period between trials. After a 1-min rest, the muscle group on the other side was measured. The highest values obtained from the two valid measurements were recorded [14].

Statistical analysis was performed using SPSS version 23 (IBM Corp., Armonk, NY, USA). To ensure a balanced randomization, we tested the differences between the RAGT group and the CON group at baseline using the Fisher’s exact test for sex and burn type, using the Mann–Whitney test for the duration between injury and initiation of training; VAS; FAC; 6MWT; isometric strengths of left ankle dorsiflexor, right ankle plantarflexor; and ROMs of bilateral hip flexion, left knee flexion, bilateral knee extension, bilateral ankle dorsiflexion, bilateral ankle plantarflexion. The values were presented as mean ± standard deviation. Intergroup comparisons were analyzed using an independent *t*-test or the Mann–Whitney test for the change scores at baseline and 12 weeks after the normality test. Pre-treatment scores were compared with post-treatment scores using the Wilcoxon signed-rank sum test and paired *t*-test after the normality test, with a significance level of *p* < 0.05.

## 3. Results

A total of 40 patients were evaluated in this study, and they underwent 60 sessions of RAGT and conventional training. The demographic characteristics of these patients are shown in Table 1. No differences were observed in the demographic and baseline clinical data between the groups. VAS scores decreased significantly from 8.06 ± 0.66 points before RAGT to 4.41 ± 1.18 points 12 weeks after RAGT (*p* < 0.001). FAC scores increased significantly from 1.76 ± 0.56 points to 4.18 ± 0.39 points 12 weeks after RAGT (*p* < 0.001). The 6MWT scores increased significantly from 204.41 ± 85.60 points to 298.53 ± 47.75 points 12 weeks after RAGT (*p* < 0.001) (Table 2). The results of the VAS, FAC, and 6MWT (8.00 ± 1.21, 1.75 ± 0.58, and 220.94 ± 116.88) before training in the CON group improved significantly (5.00 ± 1.03, 3.81 ± 1.05, and 272.19 ± 110.14) after training (*p* < 0.001, *p* < 0.001, and *p* = 0.05) (Table 2). The isometric strengths of the right hip extensor (*p* = 0.02), bilateral knee flexor (*p* = 0.04 in the right, and *p* = 0.001 in the left), bilateral knee extensor (*p* = 0.003 in the right, and *p* = 0.002 in the left), bilateral ankle dorsiflexor (*p* = 0.04 in the right, and *p* = 0.02 in the left), and bilateral ankle plantarflexor (*p* = 0.001 in the right, and *p* = 0.008 in the left) after training were significantly improved compared with those before training in the RAGT group (Table 2). The ROMs of the right knee extension (*p* = 0.03), bilateral ankle plantarflexion (*p* = 0.008 in the right, and *p* = 0.03 in the left) were significantly improved compared with measurements before training in the RAGT. There were no significant differences of the isometric strengths and ROMs of the bilateral hip, knee, and ankle muscles after training in the CON group (Table 2).

There were differences in the improvement of results of the VAS, FAC, and 6MWT between groups after training, but they were not statistically significant (*p* = 0.23, *p* = 0.14, and *p* = 0.05) (Table 3). There were significant differences in the improvements in the isometric strengths of left knee flexor (*p* = 0.01), left ankle dorsiflexor (*p* = 0.01), and left ankle plantarflexor (*p* = 0.003) between the two groups (Table 3). None of the patients experienced adverse events such as skin abrasions or worsening of joint pain during training. In addition, no surgery-related adverse events were reported.

## 4. Discussion

To our knowledge, this is the first randomized controlled trial to report the use of robot-assisted rehabilitation after burn injury. Participants tolerated the RAGT and were able to complete the training without premature withdrawal or non-compliance. Our results showed the effectiveness of RAGT in burn patients during the early rehabilitation period. Recent reports have also shown that RAGT requires less energy consumption than conventional training in deconditioned patients [16]. Therefore, RAGT can be performed for a longer time compared with a conventional gait training [17]. We proved that RAGT was safe and feasible for patients with burn injury.

The results of this study showed significant improvements in VAS, FAC, and 6MWT after intervention in both groups. Richard et al. reported that minimal clinically important differences (MCID) of 14.0–30.5 m in the 6MWT and MCID can be considered as clinically meaningful [1,18]. For RAGT, it is of critical importance to train repetitively in a natural gait similar to overground gait with proprioceptive and exteroceptive feedback [19]. RAGT can be used to provide inpatients with an intensive repetitive program of gait cycles. The robot machines can be used to assist patients to practice up to 1000 steps per session. A robot task-specific repetitive approach is regarded as the most promising method to restore gait function [20]. The intense training of the impaired limb and the acquisition of new motor skills mediate functional recovery through the sprouting of new synapses [5]. Enhancement of trunk muscle activity during robot training may lead to improvement in balance control [21]. Improved balance control from standing to sitting shows an improved gait function.

Isometric measurements (right hip extensor, bilateral knee flexor, bilateral knee extensors, bilateral ankle dorsiflexors, and bilateral ankle plantar flexors) and ROMs (right knee extension and bilateral ankle plantar flexion) after training were significantly improved compared with the measurements before training in the RAGT group. Statistically significant changes in isometric strengths of knee flexor, ankle dorsiflexor, and ankle plantarflexor were observed in the RAGT group compared with measures of CON group. Sensory deficits, muscle weakness, abnormal muscle activities, proprioception impairments, and soft tissue tightness can result in decreased gait performance [22]. Goto et al. explained that there was an improvement in the knee ROM after a robot-assisted training as shown by a resolution in quadriceps arthrogenic muscle inhibition. Moreover, the use of robot-assisted training significantly decreased the pain level during active movement and improved performance compared with conventional therapy [10]. The robot-assisted training improved the neuromuscular functions of the quadriceps, which allowed the knee to extend fully and did not increase the level of knee pain immediately after surgery [23]. Improvements in isometric strengths after RAGT in the hip, knee, and ankle were similar to the measurements after conventional strengthening program [24]. In other studies, patients who underwent RAGT demonstrated improvements in ROM of ankle dorsiflexion, isometric strength of ankle dorsiflexion, and balance scale [13,25,26]. RAGT improved the gait performance, which was indicated by an increase in muscle strength [27]. RAGT was recommended to improve the ankle performance (ankle strength, ankle range of motion, and ankle motor control) [12]. The control strategy used for ankle rehabilitation by RAGT is to assist the participants as much as possible according to real-time ankle performance. Lin et al. reported that the dorsiflexor strength was the most important factor for gait speed [22]. Patients with impaired ankle control who underwent a rehabilitation training with feedback-controlled stretching using a robot showed significant functional improvements [25]. RAGT with functional electrical stimulation on the ankle dorsiflexor of the affected limb improved the patient’s gait-related functions compared with the RAGT only [28]. This study showed improvements, which are consistent with those reported in a previous study investigating the effectiveness of RAGT. A previous study showed that stretching using a robot can improve the function of joints with impaired biomechanical properties [25]. In this study, the use of RAGT may facilitate gait functional recovery by strengthening muscle power.

Motor skills refer to the perceptible conditioned motor reflex composed of one action followed by another, including automatization. These links are correlated and gradually transit to the next action by the formation of a conditioned motor reflex. Recent studies recommended that recovering automaticity might improve function and reduce the fall risk [29]. Proprioception can be defined as the function to determine the joint as well as body movement. It is based on sensory signals provided to the brain from the skin, muscle, and joint receptors. A diminished proprioceptive acuity is associated with burn injury of the lower extremities. RAGT allows the stimulation of a normal gait cycle to reinforce the sensory inputs, and thus promotes the functional recovery [20]. The periodical movements of the knee joint during RAGT may activate the proprioceptors of the residual soft tissues around the knee joint, thus increasing the patient’s proprioception [14,20,30]. Previous studies conducted in patients with spinal cord injury showed that motor potential can be elicited by passive movement of the lower limbs by activating the gait centers in the spinal cord [31]. RAGT is believed to have satisfied the necessary external requirements for the reformation of motor skills. The integration of sensorimotor information during training enables the patient to acquire the accurate motor skill [32]. For evaluating the mechanisms of RAGT in burn patients with sensory disturbance, further studies into the changes in proprioception are needed.

This study has a limitation. It was only conducted in one center. This might have a limitation on the generalizability of the results. This study used a small sample size, which reduced the strength of the statistical power. Further works are recommended to increase patient number and centers to use RAGT and establish a relationship with number of reconstructive surgeries for early rehabilitation. Hence, future studies evaluating individuals who are unable to walk independently, examining the effect of variations in the time from injury to treatment, or in those with disturbances in proprioception may provide insights into the usefulness of robotics. Techniques for selecting important training parameters such as gait speed and step length have not been established. Additionally, further study should be conducted on RAGT over an extended period using a variety of exercise intensities.

## 5. Conclusions

The results of this study showed that RAGT is more effective than conventional gait training. This is the first study to demonstrate the effectiveness of RAGT in burn patients in terms of improving the ROMs of the lower extremities, isometric muscle strengths, and gait performance. Furthermore, RAGT is feasible and promising for application in burn physiotherapy. These results can be used as a basis when conducting RAGT in patients with burn injuries.

## Figures and Tables

**Figure 1 jcm-09-02813-f001:**
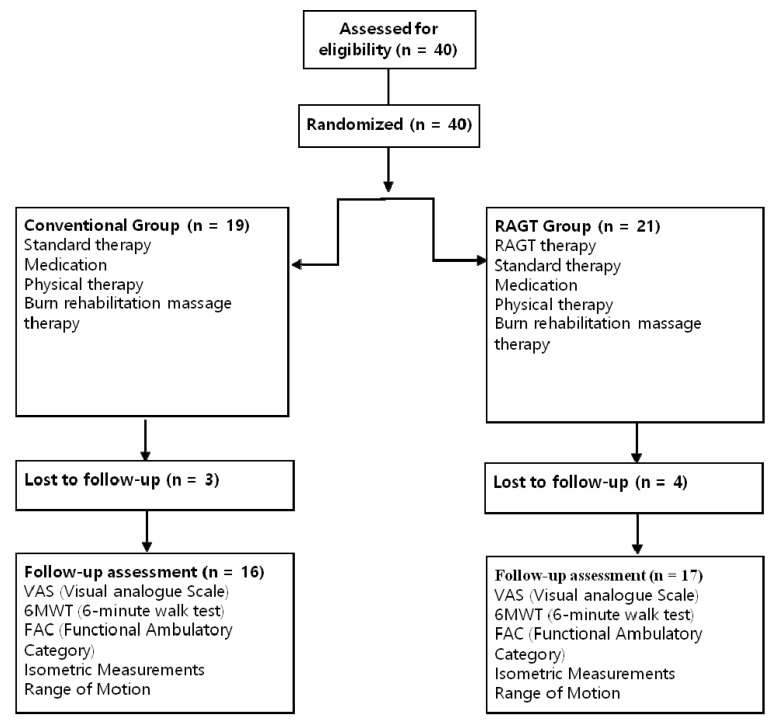
Consolidated standard of reporting trials flow diagram.

**Figure 2 jcm-09-02813-f002:**
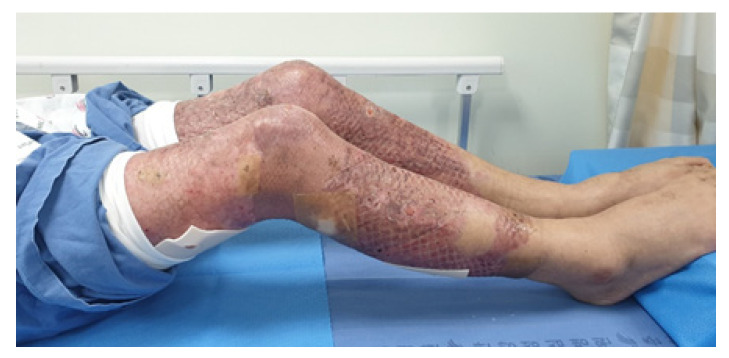
Eligible participant with a burn injury in lower extremity.

**Figure 3 jcm-09-02813-f003:**
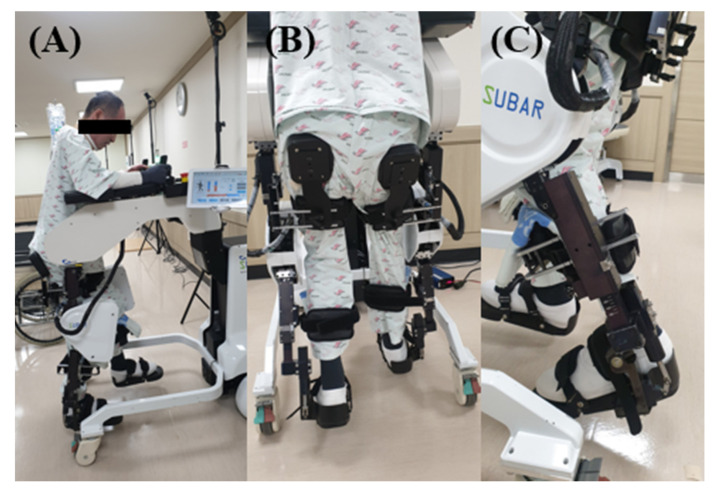
(**A**) SUBAR^®^ (CRETEM, Korea), an exoskeletal-type gait training robot; (**B**) a patient participating in gait training using the SUBAR^®^ from behind; (**C**) foot plate.

**Figure 4 jcm-09-02813-f004:**
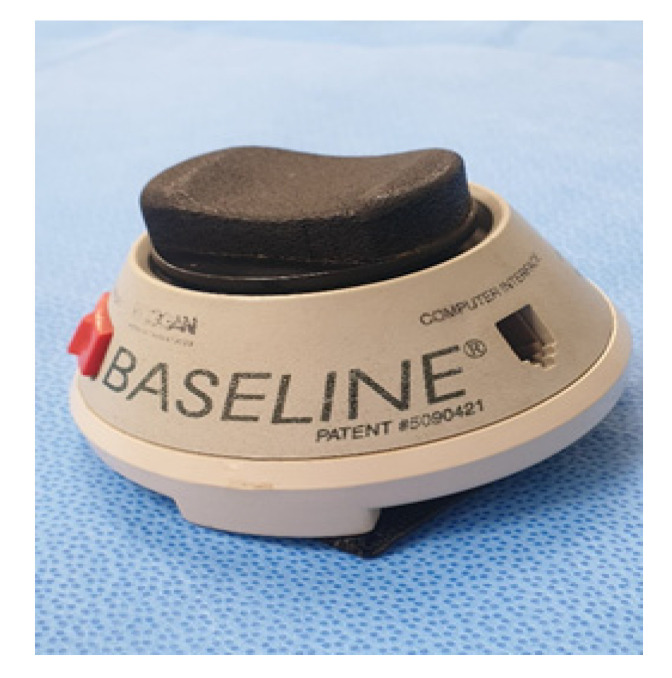
MicroFET II^TM^ (Hoggan Health Industries, Draper, UT, USA).

**Table 1 jcm-09-02813-t001:** Patients’ demographic data.

	Robot Training (*n* = 17)	Conventional Training (*n* = 16)	*p*
**Male:female**	14:3	13:3	0.64
**Age (years)**	53. 94 ± 9.51	49.06 ± 15.11	0.49
**TBSA (%)**	33.94 ± 14.64	23.19 ± 14.50	0.13
**Mechanism of burn, ** ***n*** **FB:EB:SB:CB**	9:3:4:1	7:3:2:4	0.37
**Duration (days) between burn injury and therapy**	82.29 ± 31.50	74.19 ± 44.75	0.07
**VAS**	8.06 ± 0.66	8.00 ± 1.21	0.93
**FAC**	1.76 ± 0.56	1.75 ± 0.58	0.96
**6MWT (m)**	204.41 ± 85.60	220.44 ± 10.90	0.61
**Isometric Measurements (Nm)**			
Hip flexor, right	22.71 ± 4.79	28.44 ± 10.90	0.08
Hip flexor, left	23.59 ± 7.87	26.69 ± 11.75	0.38
Hip extensor, right	16.41 ± 4.21	18.38 ± 7.46	0.36
Hip extensor, left	18.29 ± 7.86	15.81 ± 5.95	0.32
Knee flexor, right	15.94 ± 3.03	17.63 ± 2.92	0.12
Knee flexor, left	15.12 ± 4.54	16.06 ± 5.23	0.58
Knee extensor, right	17.00 ± 6.15	20.38 ± 6.34	0.13
Knee extensor, left	16.71 ± 8.47	21.13 ± 7.14	0.12
Ankle dorsiflexor, right	15.06 ± 8.42	17.75 ± 7.39	0.34
Ankle dorsiflexor, left	14.01 ± 6.26	14.50 ± 8.69	0.99
Ankle plantarflexor, right	15.21 ± 8.84	18.06 ± 7.22	0.14
Ankle plantarflexor, left	16.21 ± 8.05	17.19 ± 8.56	0.74
**Range of Motion (degree)**			
Hip flexion, right	97.06 ± 6.14	100.00 ± 3.65	0.28
Hip flexion, left	95.00 ± 11.18	98.75 ± 5.00	0.40
Hip extension, right	16.94 ± 5.92	19.75 ± 6.07	0.19
Hip extension, left	18.35 ± 6.10	19.75 ± 6.07	0.23
Knee flexion, right	119.41 ± 16.67	126.19 ± 18.28	0.27
Knee flexion, left	111.71 ± 27.12	122.00 ± 31.98	0.14
Knee extension, right	−5.94 ± 8.93	0.13 ± 2.99	0.05
Knee extension, left	−2.47 ± 5.39	−0.81 ± 3.15	0.09
Ankle dorsiflexion, right	16.29 ± 8.04	18.31 ± 7.94	0.51
Ankle dorsiflexion, left	15.82 ± 6.28	14.19 ± 7.34	0.71
Ankle plantarflexion, right	28.35 ± 11.89	35.69 ± 8.73	0.12
Ankle plantarflexion, left	29.88 ± 11.48	30.56 ± 11.28	0.85

TBSA: total body surface area; FB: flame burn; EB: electrical burn; SB: scalding burn; CB: contact burn; VAS: visual analogue scale; FAC: functional ambulatory category; 6MWT: 6-min walking test.

**Table 2 jcm-09-02813-t002:** Comparison between the scores before training and after training.

	Robot Training (*n* = 17)		*p*	Conventional Training (*n* = 16)		*p*
	Before Training	After Training		Before Training	After Training	
**VAS**	8.06 ± 0.66	4.41 ± 1.18	<0.001 *	8.00 ± 1.21	5.00 ± 1.03	<0.001 *
**FAC**	1.76 ± 0.56	4.18 ± 0.39	<0.001 **	1.75 ± 0.58	3.81 ± 1.05	<0.001 **
**6MWT**	204.41 ± 85.60	298.53 ± 47.75	<0.001 *	220.94 ± 116.88	272.19 ± 110.14	0.005 *
**Isometric measurements (Nm)**						
Hip flexor, right	22.71 ± 6.79	25.24 ± 7.55	0.08	28.44 ± 10.90	29.38 ± 10.28	0.40
Hip flexor, left	23.59 ± 7.87	25.00 ± 7.24	0.32	26.69 ± 11.75	27.31 ± 11.88	0.49
Hip extensor, right	16.41 ± 4.21	18.24 ± 5.47	0.02 *	18.38 ± 7.46	18.81 ± 6.91	0.68
Hip extensor, left	18.29 ± 7.86	18.59 ± 6.43	0.84	15.81 ± 5.95	16.00 ± 5.91	0.80
Knee flexor, right	15.94 ± 3.03	17.71 ± 5.92	0.04 **	17.63 ± 2.92	18.94 ± 6.06	0.34
Knee flexor, left	15.12 ± 4.54	19.76 ± 5.30	0.001 *	16.06 ± 5.23	16.13 ± 6.96	0.96
Knee extensor, right	17.00 ± 6.15	23.00 ± 6.21	0.003 *	20.38 ± 6.34	25.19 ± 9.17	0.04 *
Knee extensor, left	16.71 ± 8.47	23.29 ± 7.19	0.002 *	21.13 ± 7.14	21.25 ± 8.62	0.93
Ankle dorsiflexor, right	15.06 ± 8.42	17.71 ± 8.45	0.04 **	17.75 ± 7.39	18.44 ± 6.67	0.89
Ankle dorsiflexor, left	14.01 ± 6.26	17.24 ± 7.94	0.02 **	14.50 ± 8.69	14.19 ± 8.38	0.79
Ankle plantarflexor, right	15.21 ± 8.84	20.35 ± 8.90	0.001 **	18.06 ± 7.22	19.63 ± 7.08	0.18
Ankle plantarflexor, left	16.21 ± 8.05	20.88 ± 9.16	0.008 *	17.19 ± 8.56	16.06 ± 6.55	0.22
**Range of motion (degree)**						
Hip flexion, right	97.06 ± 6.14	99.76 ± 0.97	0.07	100.00 ± 3.65	99.63 ± 1.50	0.66
Hip flexion, left	95.00 ± 11.18	99.76 ± 0.97	0.07	98.75 ± 5.00	99.50 ± 2.00	0.32
Hip extension, right	16.94 ± 5.92	17.94 ± 6.65	0.27	19.75 ± 6.07	21.25 ± 7.02	0.78
Hip extension, left	18.35 ± 6.10	19.94 ± 5.13	0.16	20.63 ± 4.50	20.13 ± 9.46	0.81
Knee flexion, right	119.41 ± 16.67	115.88 ± 20.21	0.26	126.19 ± 18.28	132.63 ± 16.32	0.22
Knee flexion, left	111.71 ± 27.12	117.47 ± 19.65	0.21	122.00 ± 31.98	121.44 ± 25.51	0.88
Knee extension, right	−5.94 ± 8.93	−1.18 ± 3.64	0.03 **	0.13 ± 2.99	−0.19 ± 0.75	0.66
Knee extension, left	−2.47 ± 5.39	−1.12 ± 2.03	0.12	−0.81 ± 3.15	−0.69 ± 1.54	0.66
Ankle dorsiflexion, right	16.29 ± 8.04	16.65 ± 5.93	0.44	18.31 ± 7.94	16.88 ± 7.27	0.53
Ankle dorsiflexion, left	15.82 ± 6.28	16.59 ± 6.01	0.34	14.19 ± 7.34	13.19 ± 11.09	0.40
Ankle plantarflexion, right	28.35 ± 11.89	36.18 ± 8.00	0.008 **	35.69 ± 8.73	39.63 ± 1.50	0.14
Ankle plantarflexion, left	29.88 ± 11.48	36.12 ± 7.70	0.03 **	30.56 ± 11.28	34.63 ± 7.56	0.09

VAS: visual analogue scale; FAC: functional ambulatory category; 6MWT: 6-min walking test; * *p* < 0.05 paired *t*-test; pre- and post-intervention measurements were compared; ** *p* < 0.05 Wilcoxon signed-rank test; pre- and post-intervention measurements were compared.

**Table 3 jcm-09-02813-t003:** Change scores (before and after training) on measured outcomes.

	Robot Training (*n* = 17)	Conventional Training (*n* = 16)	*p*
**VAS**	−3.65 ± 1.50	−3.00 ± 1.51	0.23
**FAC**	2.41 ± 0.62	2.06 ± 0.77	0.14
**6MWT**	94.12 ± 61.23	51.25 ± 61.55	0.05
**Isometric measurements (Nm)**			
Hip flexor, right	2.53 ± 5.60	0.94 ± 3.62	0.23
Hip flexor, left	1.41 ± 5.62	0.63 ± 3.50	0.64
Hip Extensor, right	1.82 ± 3.00	0.44 ± 2.66	0.13
Hip extensor, left	0.29 ± 4.55	0.19 ± 2.97	0.94
Knee flexor, right	1.76 ± 5.24	1.31 ± 5.34	0.81
Knee flexor, left	4.65 ± 4.61	0.06 ± 4.45	0.01 *
Knee extensor, right	6.00 ± 6.91	4.81 ± 8.79	0.67
Knee extensor, left	6.59 ± 7.13	0.13 ± 5.66	0.01 *
Ankle dorsiflexor, right	2.65 ± 4.39	0.69 ± 6.01	0.14
Ankle dorsiflexor, left	3.22 ± 4.69	−0.31 ± 4.53	0.02 **
Ankle plantarflexor, right	5.15 ± 6.22	1.56 ± 5.32	0.09
Ankle plantarflexor, left	4.67± 6.35	−1.13 ± 3.52	0.003 *
**Range of motion (degree)**			
Hip flexion, right	2.71 ± 5.71	−0.38 ± 2.75	0.26
Hip flexion, left	4.76 ± 10.77	0.75 ± 3.00	0.40
Hip extension, right	1.00 ± 5.74	1.50 ± 5.98	0.33
Hip extension, left	1.59 ± 4.39	−0.50 ± 9.45	0.33
Knee flexion, right	−3.53 ± 12.50	6.44 ± 19.98	0.09
Knee flexion, left	5.76 ± 20.44	−0.56 ± 14.23	0.31
Knee extension, right	4.76 ± 7.78	−0.31 ± 2.87	0.09
Knee extension, left	1.35 ± 4.33	0.13 ± 3.30	0.14
Ankle dorsiflexion, right	0.35 ± 7.07	−1.44 ± 7.08	0.38
Ankle dorsiflexion, left	0.76 ± 2.99	−1.00 ± 6.16	0.33
Ankle plantarflexion, right	7.82 ± 10.24	3.94 ± 8.87	0.10
Ankle plantarflexion, left	6.24 ± 9.90	4.06 ± 8.87	0.85

VAS: visual analogue scale; FAC: functional ambulatory category; 6MWT: 6-min walking test, * *p* < 0.05 independent *t*-test; ** *p* < 0.05 Mann–Whitney test.

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
