# Peer review of "Effects of Robot-Assisted Gait Training in Patients with Burn Injury on Lower Extremity: A Single-Blind, Randomized Controlled Trial"

_jcm, 2020, doi:10.3390/jcm9092813_

Round 1

Reviewer 1 Report

Can't find my previous comments, read through it again, and seems to be reasonable.

Reviewer 2 Report

A very nice read. Good for publication.

This manuscript is a resubmission of an earlier submission. The following is a list of the peer review reports and author responses from that submission.

Round 1

Reviewer 1 Report

Very important study. Future work: increase patient number and centers to use RAGT and establish a relationship with number of reconstructive surgeries perform in the patients with early RAGT rehabilitation.

Contracture is the major complication after burn injury. Early rehabilitation after burn injury may improve outcome. RAGT facilitate early recovery and it is feasible and promising.

Author Response

Very important study. Future work: increase patient number and centers to use RAGT and establish a relationship with number of reconstructive surgeries perform in the patients with early RAGT rehabilitation.

Contracture is the major complication after burn injury. Early rehabilitation after burn injury may improve outcome. RAGT facilitate early recovery and it is feasible and promising.

Answer> We appreciate you careful advise. We mentioned about future work in the discussion section.

Reviewer 2 Report

*  The description in the abstract is not clear.  How much time was spent in each group>  It looks like the robot group had more time in therapy?

*  Visual Analog Scale of what, pain?

*  Significant improvement in VAS, FAC, and six-,minute walk, how much?

*  Hypotheses are generally clear statements to test; please refrain from hedging by saying 'may', or 'might'.

*  Please use the past tense in the Methods

*  Please include full-thickness total body surface area burned as well in the baseline characteristics.

*   It is uncommon for all persons who are approached for enrollment to actually participate.  In yours, this was 100% which is likely not true.  Please confirm.  Further, in the CONSORT diagramme, please include an accounting of all who were eligible for the reader to assess potential sources of bias.

*  How was randomization achieved?  By what method?

*  In the Methods, please be clearer on the FAC number; was it <1 or <3?

*  For the unstable fractures, do you mean weight-bearing restrictions?  

*  What is a skin disorder that is worsened by RAGT?  Healing burn wound for instance?

*  Please define 'fixed' contracture.  In the first 2 years after injury, no burn scar contractures are 'fixed' but respond to therapies in general.

*  For the CON group, what did they do for the other 30 minutes per session since they received 60 minute sessions? The Methods unfortunately are not clear,...

*  How exactly did the subjects report VAS?  Was this a paper form, or were they asked to provide a number verbally?  Unfortunately, the latter is a source of bias since it was being recorded by an unblinded observer, and oftentimes subjects will report whatever the recorder wants to hear

*  For the analysis, I am having a hard time determining whether an advantage was present in the RAGT group; both groups improved, but was RAGT even more improved for VAS, FAC, and six-minute walk?

*  The results on ROM And strength are interesting!  However, we also need to see a between groups comparison here too. 

Author Response

Thank you for the encouraging reviews of our paper entitled, “Effects of robot-assisted gait training in patients with burn injury on lower extremity: a single blind, randomized controlled trial

”We responded to the reviewers’ comments as follows:

Please find our response to the reviewers’ comments below.

  1. The description in the abstract is not clear. How much time was spent in each group>  It looks like the robot group had more time in therapy?, Visual Analog Scale of what, pain?, Significant improvement in VAS, FAC, and six-,minute walk, how much?,

Answer> We agree with the reviewer. We added the descriptions in the abstract section.

  1. Hypotheses are generally clear statements to test; please refrain from hedging by saying 'may', or 'might'.

Answer> We agree with the reviewer. We rephrased the sentences in the abstract section.

  1. Please use the past tense in the Methods

Answer> We agree with the reviewer. We revised the sentences in the method section.

  1. Please include full-thickness total body surface area burned as well in the baseline characteristics., In the Methods, please be clearer on the FAC number; was it <1 or <3?, For the unstable fractures, do you mean weight-bearing restrictions?, What is a skin disorder that is worsened by RAGT?  Healing burn wound for instance?, Please define 'fixed' contracture.  In the first 2 years after injury, no burn scar contractures are 'fixed' but respond to therapies in general.

Answer> We agree with the reviewer. For detailed information, we revised the inclusion criterias in the method section. We hope this will help the reader to understand with ease.

  1. It is uncommon for all persons who are approached for enrollment to actually participate.  In yours, this was 100% which is likely not true.  Please confirm.  Further, in the CONSORT diagramme, please include an accounting of all who were eligible for the reader to assess potential sources of bias. How was randomization achieved?  By what method?

Answer> We appreciate you careful advise. We added the detailed descriptions of randomization and enrollmnet in the mothod section.

  1. For the CON group, what did they do for the other 30 minutes per session since they received 60 minute sessions? The Methods unfortunately are not clear.

Answer> We agree with the reviewer. We added the explanation of treatment time in the method section.

  1. How exactly did the subjects report VAS?  Was this a paper form, or were they asked to provide a number verbally?  Unfortunately, the latter is a source of bias since it was being recorded by an unblinded observer, and oftentimes subjects will report whatever the recorder wants to hear.

Answer> We appreciate you careful advise. We added detailed explanation about evaluation.

  1. For the analysis, I am having a hard time determining whether an advantage was present in the RAGT group; both groups improved, but was RAGT even more improved for VAS, FAC, and six-minute walk?, The results on ROM And strength are interesting!  However, we also need to see a between groups comparison here too. 

Answer> We agree with the reviewer. We added the comparisons between groups after training in the results section.

Reviewer 3 Report

Overall good study. Would be nice to see this in larger population. Any comments on Obese patients (which is a major problem in US) in terms of weight limitation for robot use, fitting???, how about cost vs conventional therapy. Can non academic institution afford/incorporate this in their rehab center? Helpful to know the extent of burn depths, i.e full thickness vs partial thickness etc..  My other recommendations as follow:

line 19- need a space after motion (ROM) 

line 43- Rehabilitation has been an essential for improving the gait and independence.. no need to mention burn patients here since this paper is about them.

line 45-46- If the patients were able to stand, they were assisted to ambulate progressively and even start gait training.

line 54-55- In the present randomised controlled trial, we aimed to verify whether the RAGT can be used in rehabilitation for burn patients to promote functional recovery.

line 174- remove AND. Statistical changes..

line 185-186- In other studies, patients who underwent RAGT demonstrated improvements in ROM of ankle dorsiflexion, isometric strength of ankle dorsiflexion, and balance 186 scale [22-24].

line 214-216- please rephrase this sentence.

Author Response

Thank you for the encouraging reviews of our paper entitled, “Effects of robot-assisted gait training in patients with burn injury on lower extremity: a single blind, randomized controlled trial

”We responded to the reviewers’ comments as follows:

Please find our response to the reviewers’ comments below.

1.Would be nice to see this in larger population.

Answer> We agree with the reviewer. We added the limitation in the discussion section.

2. Any comments on Obese patients (which is a major problem in US) in terms of weight limitation for robot use, fitting???,

Answer> We appreciate you careful advise. Due to the belt length for fixing the thigh and calf, the weight limit for riding the robot is 100kg. We added the descriptions in the method section.

3. how about cost vs conventional therapy.  

Answer> We appreciate you careful advise. We added the explanation of the cost in the method section.

4. Can non academic institution afford/incorporate this in their rehab center?

Answer> This study was carried out by rehabilitation robots supplied from the Ministry of Health & Welfare and the National Rehabilitation Center in 2018. But this study was supported by Translational Research Program for Rehabilitation Robots (NRCTR-EX19002). We added the funding source.

5. Helpful to know the extent of burn depths, i.e full thickness vs partial thickness etc.

Answer> We appreciate you careful advise. For detailed information, we revised the inclusion criterias in the method section. We hope this will help the reader to understand with ease.

6. line 19- need a space after motion (ROM), line 43- Rehabilitation has been an essential for improving the gait and independence.. no need to mention burn patients here since this paper is about them., line 45-46- If the patients were able to stand, they were assisted to ambulate progressively and even start gait training., line 54-55- In the present randomised controlled trial, we aimed to verify whether the RAGT can be used in rehabilitation for burn patients to promote functional recovery.  

Answer> We agree with the reviewer. We rephrased the sentences as pointed out in the background section.

7. line 174- remove AND. Statistical changes., line 185-186- In other studies, patients who underwent RAGT demonstrated improvements in ROM of ankle dorsiflexion, isometric strength of ankle dorsiflexion, and balance scale [22-24]., line 214-216- please rephrase this sentence.

Answer> We agree with the reviewer. We rephrased the sentences as pointed out in the discussion section.

Round 2

Reviewer 2 Report

*  Unfortunately, in requesting an additional analysis, the authors showed no significant effect of the treatment other than some changes in muscle strength, which was probably related to power rather than the treatment itself.  Therefore, more subjects are needed to support the conclusions and potential impact of the paper, and thus the report is premature.  Therefore, I now recommend reject and resubmit with more subjects.

For the other issues:

*  The hypothesis for the study stated in the Introduction is still vague and non-specific.

*  The CONSORT diagramme is still incomplete.  How many were available for screening?  All 28 potential subjects agreed to participate?

*  It is still not clear how subjects were recruited and randomised.  In general, in a reply to referees, it is beneficial to state exactly where the changes were made rather than using generalities such 'the Methods section'.

*  How the VAS scores were collected and recorded is still not clear.

Author Response

Thank you for the encouraging reviews of our paper entitled, “Effects of robot-assisted gait training in patients with burn injury on lower extremity: a single blind, randomized controlled trial

”We responded to the reviewers’ comments as follows:

Please find our response to the reviewers’ comments below.

Reviewer #2:

  1. Unfortunately, in requesting an additional analysis, the authors showed no significant effect of the treatment other than some changes in muscle strength, which was probably related to power rather than the treatment itself. 

Answer> We appreciate you careful advise. It was known that the ankle and knee joint contracture aften developed after burns, and the ankle and knee had severe degrees of joint contractures[1]. Therefore, it was thought that some improvement of the ankle and knee joint was a meaningful result.

[1] Tan J, Chen J, Zhou J, Song H, Deng H, Ao M, et al. Joint contractures in severe burn patients with early rehabilitation intervention in one of the largest burn intensive care unit in China: a descriptive analysis. Burns Trauma. 2019;7:17.

  1. Therefore, more subjects are needed to support the conclusions and potential impact of the paper, and thus the report is premature.  Therefore, I now recommend reject and resubmit with more subjects.

Answer> We agree with the reviewer. So far, research on robot rehabilitation for musculoskeltal patients is in its infancy. As you can see form the added references there are not many experimental patients[2]. Among the musculoskeletal patients, burn patients belong to a small group, and in this study, a small number of experimental patients belong to the limitations. But this study is consided to be meaningful in the sense that robot rehabilitation is first applied to burn patients. As the reviewer pointed out, we plan to study more patients in the future.

[2] Zhang M, Davies TC, Xie S. Effectiveness of robot-assisted therapy on ankle rehabilitation--a systematic review. J Neuroeng Rehabil. 2013;10:30.

  1. The hypothesis for the study stated in the Introduction is still vague and non-specific.

Answer> We appreciate you careful advise. We explained the first attempt of robot rehabilitation in patients with burns and the justification for evaluating the effects in the introduction. We added more references in the introduction section.

  1. The CONSORT diagramme is still incomplete.  How many were available for screening?  It is still not clear how subjects were recruited and randomised. All 28 potential subjects agreed to participate?

Answer> We appreciate you careful advise. We added the detailed descriptions of randomization and enrollmnet in the mothod section. All 28 patients were provided a written informed consent. We added the descriptions of the patients who dropped out in this study.

Numbers were assigned to 28 burn patients according to the order of admission who satisfied all the aforementioned criteria. A computer program was used to randomly divide them into the RAGT group (n=14) or the CON group (n=14). Four patients in the RAGT group and three patients in the CON group dropped out of the study because they did not want to undergo serial evaluations after recovering the gait function and did not visit the outpatient clinic for gait training.

  1. How the VAS scores were collected and recorded is still not clear.

Answer> We appreciate you careful advise. The evaluation method was further explained. 

Visual analogue scale (VAS) was used to rate the degree of subjective pain during gait movement: 0 points were assigned when no pain was noted, and unbearable pain was assigned 10 points. VAS in gait was measured using a questionnaire.